# ACCELERATING SIMULATION-BASED INFLUENCE MAXIMIZATION VIA BAYESIAN OPTIMIZATION

## ABSTRACT

Influence Maximization (IM) has garnered significant attention due to its broad applicability in areas such as viral marketing, social network recommendations, and disease containment. The primary goal of IM is to identify an optimal seed set that maximizes influence spread. Existing methodologies for IM are largely categorized into proxy-based and simulation-based approaches, each with its own limitations. Proxy-based methods often fail to capture complex seed interactions and are model-specific, while simulation-based techniques are computationally expensive for large-scale graphs. Additionally, current research lacks a comprehensive model to understand the relationship between seed set configurations and their resulting influence spreads. To address these challenges, we present a Bayesian Optimization Influence Maximization (**BOIM**) framework that employs Bayesian optimization to minimize the number of required simulations. Our approach utilizes a Gaussian Process (GP) as the surrogate function to model the complex interplay between seed sets and influence spreads. In GP, we also introduce a specialized kernel for graph-level Bayesian optimization and implement stratified sampling to ensure uniform instance distribution. Our methodology offers a computationally efficient yet accurate alternative to traditional IM approaches, effectively bridging the gap between computational efficiency and approximation fidelity. Extensive experimentation has demonstrated that our approach has effectiveness and efficiency that surpasses standard simulation methods.

## 1 INTRODUCTION

In an increasingly networked world, the concept of Influence Maximization (IM) has risen to prominence, attracting sustained interest from both the academic and industrial communities Domingos & Richardson (2001); Li et al. (2018). This significance is underscored by its wide-ranging applications, including but not limited to, viral marketing Chen et al. (2010), personalized recommendations in social networks Ye et al. (2012), rumor mitigation He et al. (2012), and the containment of infectious diseases Newman (2002). The core objective of IM is to identify a seed set of size $k$ that optimizes the extent of influence propagation, commonly referred to as influence spread. Given that solving IM problems optimally is NP-hard, existing research often resorts to approximation techniques, primarily greedy algorithms Kempe et al. (2003); Tong et al. (2010); Yan et al. (2019); Zhang et al. (2022). Nevertheless, the suboptimal performance or computational inefficiency of current IM algorithms can lead to significant repercussions. As such, the timely identification of the most effective seed set remains a research imperative.

Current research on IM faces two major pain points: **(1) Formulation of an Optimal Algorithm for Influence Maximization:** Influence Maximization (IM) methodologies predominantly bifurcate into two paradigms: proxy-based and simulation-based approaches. Proxy-based techniques emphasize computational efficiency and have evolved over time. Initial methods employed heuristic estimations of nodal influence Chen et al. (2009); Roth (1988); Chung et al. (2003), while subsequent advancements have strived for a more nuanced capture of propagation dynamics Yan et al. (2019); Zhang et al. (2022). Despite their computational advantages, these methods often exhibit limitations in capturing intricate seed interactions and are highly model-specific, thereby compromising approximation quality. Specifically, they fall short in identifying overlapping influence flows, leading to suboptimal results. Conversely, simulation-based methods prioritize approximation fidelity by iteratively selecting nodes that maximize marginal influence Kempe et al. (2003). However, the

computational burden escalates exponentially for large-scale graphs, thereby impeding practical applicability Arora et al. (2017). Subsequent research has aimed to ameliorate this by devising more efficient simulation algorithms at the expense of some approximation accuracy Cheng et al. (2013); Borgs et al. (2014). The overarching challenge remains to accelerate simulation-based methods without sacrificing their approximation fidelity. **(2) Modeling the Interplay between Seed Set and Influence Propagation:** The prevailing focus in existing IM literature has predominantly been the identification of influential seed sets, often neglecting the intricate relationship between seed selection and resultant influence spread. Zhang et al. Zhang & Chen (2023) made an initial attempt to dissect the influence contributions of individual seeds and their interdependencies using global sensitivity analysis. However, this discourse still overlooks the latent correlations between seed set configurations and ultimate influence outcomes. Another line of inquiry employs Graph Neural Networks as a predictive model for individual seed influence Kumar et al. (2022), but this is constrained by model transferability and the computational overhead of training data generation. Moreover, the final seed selection still adheres to a greedy paradigm, thereby neglecting potential synergistic interactions among seeds. Consequently, there exists a conspicuous gap in the development of a robust methodology that can accurately model the relationship between seed set configurations and their ensuing influence spread.

To address the aforementioned challenges, we introduce a Bayesian Optimization Influence Maximization (**BOIM**), a method that leverages the sample efficiency of Bayesian optimization (BO) to significantly reduce the requisite number of simulations. Specifically, we employ a surrogate Gaussian Process function to capture the intricate relationship between seed sets and their corresponding influence spreads. To facilitate graph-level BO, we design a specialized kernel that accurately quantifies the distance between seed sets within the graph structure. Moreover, we implement a stratified sampling technique, preceded by clustering, across the graphs to ensure a uniform distribution of sampled instances within each BO iteration. This methodology not only enhances performance but also expedites the optimization process, offering a more computationally efficient alternative to traditional simulation-based approaches for IM. Our primary contributions include:

- **Propose an efficient and effective simulation-based method.** This method deviates from conventional simulation-based methods and greedy methods by incorporating the consideration of seed interactions. In addition, simulation-based algorithms allocate a substantial amount of computer resources and time to execute simulations. However, the Bayesian optimization paradigm enhances the feasibility of the algorithm by substantially decreasing the number of simulations required.

- **Provide theoretical support for the proposed kernel and sampling in IM.** To ensure the robustness and efficacy of our proposed kernel method, we rigorously validate the kernel function through theoretical analysis. Also, we provide a comprehensive theoretical framework to substantiate that our graph-based sampling approach significantly mitigates variance. This theoretical investigation serves as a robust complement to our empirical studies, thereby rendering our research both comprehensive and methodologically sound.

- **Conduct extensive empirical experiments to prove the superiority of BOIM .** To substantiate the efficacy and efficiency of the proposed algorithm, we employ an extensive suite of both real-world and synthetic datasets. Our algorithm not only attains performance metrics that are on par with traditional simulation-based methods but also exhibits a computational speedup, executing approximately 1.5 times faster than the most efficient existing algorithm.

## 2 RELATED WORK

**Influence Maximization**. AS IM is NP-hard, researchers have pursued feasible solutions with optimal performance. The first approximation approach proposed a simulation-based greedy algorithm but it lacked scalability Kempe et al. (2003). Subsequently, other simulation-based methods were developed to improve performance or reduce complexity, yet high computational costs persist, prohibiting application to massive online networks Leskovec et al. (2007); Goyal et al. (2011); Arora et al. (2017); Tang et al. (2014). Critically, simulation opacity precludes elucidating and enhancing diffusion processes Li et al. (2018). To mitigate burdensome simulations, proxy-based approaches emerged, whereby node spreading power is approximated by proxies. Initial proxies were simple

heuristics like degree, PageRank Page et al. (1999), and eigen-centrality Zhong et al. (2018). Later, influence-aware and diffusion model-aware proxies were proposed to better estimate seed influence spread Chen et al. (2009); Kimura et al. (2009); Yan et al. (2019); Zhang et al. (2022). **Diffusion models** describe how the node tries to activate its neighbors. Independent cascade (IC) and linear threshold (LT) are common examples Kempe et al. (2003). In IC, each node will try once and only once to activate all its neighbors with a certain probability in each time step. The influence spread of a seed set has an uncertainty that comes from the probabilities of the nodes activating each other. This uncertainty is not what researchers are interested in. Thus, it is often removed by taking the average of multiple simulation rounds as the expected influence spread. In LT, edges are weightless while each node has an individual threshold. When the percentage of a node's neighbors that are activated exceeds its threshold, the node will be activated in the next time step. Since the information about each node's threshold is not available, it is randomly assigned in each round of the simulation.

**Bayesian Optimization** is an approach for optimizing black-box functions that are expensive to evaluate. It constructs a probabilistic model of the objective function and uses this model to determine promising candidates to evaluate next Frazier (2018). Bayesian optimization was first proposed by Mockus et al. Mockus (1998) and has since become a popular methodology for hyperparameter tuning and optimization of complex simulations and models Snoek et al. (2012). The key idea is to leverage Bayesian probability theory to model uncertainty about the objective function. A prior distribution is placed over the space of functions, often a Gaussian process, which is updated as observations are made. An acquisition function then uses this model to determine the next evaluation point by balancing exploration and exploitation. Some common acquisition functions include expected improvement, knowledge gradient, and upper confidence bound Shahriari et al. (2015). There has been much work extending Bayesian optimization to handle constraints Gelbart et al. (2014), parallel evaluations González et al. (2016), and high dimensions de Freitas & Wang (2013). Overall, Bayesian optimization provides an elegant and principled approach to sample-efficient optimization of black-box functions. Bayesian optimization over a graph search space has emerged in the past decades. However, most of the works focus on node-level tasks and thus develop specific kernels for node smoothing Ng et al. (2018); Oh et al. (2019); Walker & Glocker (2019); Opolka & Liò (2020); Borovitskiy et al. (2021); Opolka et al. (2022). These works, while related, deal with a different task and the methods cannot be applied on our problem.

## 3 PROBLEM SETUP

A graph is represented as a bidirectional structure $G = (\mathcal{V}, \mathcal{E})$, where $\mathcal{V}$ and $\mathcal{E}$ denote the set of nodes and edges, respectively, and $|\mathcal{V}| = N$. Given this graph $G$, a predefined seed budget $k \in \mathbb{N}^+$, and a specific diffusion model $d$, the objective of an Influence Maximization (IM) algorithm is to identify a seed set $\Omega$ of size $k$ that approximately maximizes the expected influence spread $\phi(\Omega)$ (i.e., numbers of covered nodes). Mathematically, this can be formulated as:

$$\Omega = \underset{\Omega}{\operatorname{argmax}} \, \phi(\Omega), \quad \text{s.t.} \quad |\Omega| \leq k. \tag{1}$$

To capture the underlying relationship between the seed set $\Omega$ and the influence spread $\phi(\Omega)$, we aim to approximate a function $f$ such that $\phi(\Omega) \approx f(\Omega; G, d)$. Consequently, Equation 1 can be reformulated as:

$$\Omega = \underset{\Omega}{\operatorname{argmax}} \, f(\Omega; G, d), \quad \text{s.t.} \quad |\Omega| \leq k. \tag{2}$$

## 4 METHOD

This research presents a learning framework called **BOIM**, which is based on Bayesian optimization. The framework is designed for the purpose of influence maximization, as depicted in Figure 1. The first step involves the development of a graph spectral kernel (GSG) function, which serves the purpose of quantifying the similarity between sets of source nodes. By including this kernel, we utilize a Gaussian Process (GP) model to function as as surrogate for predicting the influence spread given a specified source set of nodes. The candidate source sets are organized into distinct clusters, and in order to enhance the initialization of the Gaussian Process (GP) model, we perform several simulations and utilize stratified sampling techniques on graphs. This approach aims to reduce variance during selecting the most favorable candidate nodes. The process of updating subsequent

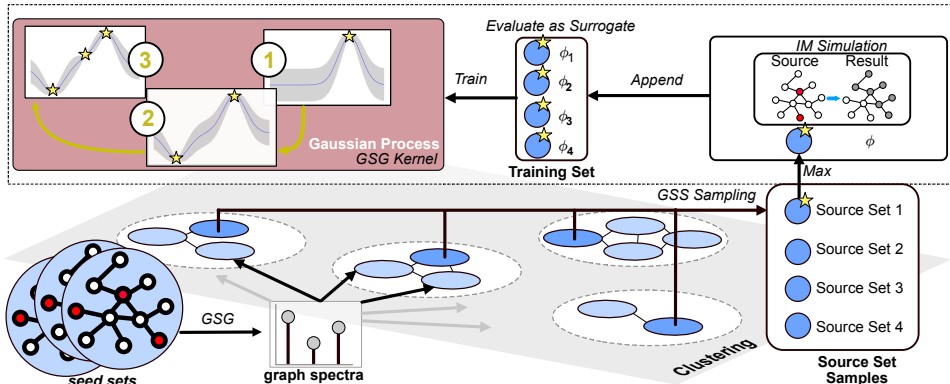

FIGURE 1: Overview of **BOIM**

models involves iteratively selecting training instances based on the Expected Improvement (EI) criterion. The aforementioned procedure will be repeated multiple times under a budget constraint. Ultimately, the complete exploration of all candidate cases leads to the identification of the ideal collection of influential nodes.

## 4.1 REDUCE SEARCH SPACE

Consider a graph with $N$ nodes, node sets are typically associated with a binary vector, where they are labeled as $1$ if being sources and $0$ otherwise. This vector is represented with $s = \{0, 1\}^N$. With $k$ sources, the total possible source configurations is $\binom{N}{k}$. Recognizing that not all nodes are equally significant in diffusion, like major cities in transport networks or key influencers in social networks, we focus on the top $a$ nodes by degree. This reduces potential source combinations to $\binom{a}{k}$ where $a \ll N$.

Past Influence Maximization (IM) research has shown that selecting seeds in close proximity to each other diminishes the final influence spread due to the overlap of their influence regions Tong et al. (2010); Zhang & Chen (2023). To mitigate this, we maximize the inner distance among a seed set $s$ by:

$$d_s = \max_s \min_{u,v} d(u, v), \quad \forall u, v \in s \tag{3}$$

where $d(u, v)$ denotes the shortest distance between two nodes. The seed sets with top $d_s$ values are selected as candidate sets.

## 4.2 KERNEL DESIGN FOR GAUSSIAN PROCESS

A kernel that is appropriate and valid ensures that Gaussian Processes (GPs) reliably estimate the extent of influence propagation, given a specific seed set. One of the issues lies in the absence of graph structural information in the binary seed vector representation $s$. To illustrate, consider two 3-node sets: one original and the other formed by shifting each node by one hop based on the original one. Although it is anticipated that the final influence spread would be quite similar for these two sets, the similarity of the binary representations is actually quite low (0 in this case). This binary representation not only inadequately characterizes the similarity between two sets of nodes, but also violates the smoothness assumption imposed by the Gaussian process. Previous work for graph kernels prioritizes structural comparisons, often ignoring attributes over the graphs Vishwanathan et al. (2010); Kriege et al. (2020); Nikolentzos et al. (2021); Siglidis et al. (2020). In order to overcome this constraint, we propose a novel kernel that effectively combines graph structure information and attributes with theoretical validity. First, the source vector $s$ is transformed into its Fourier counterpart $\tilde{s}$ such that:

$$\tilde{s} = U^\top s, \quad \tilde{s}(i) = \sum_{i=1}^{n} s_i U^\top(i), \tag{4}$$

where $U^\top$ is the inverse eigenvectors of the graph Laplacian and serves as a graph Fourier transform basis. Combining the graph Fourier transform and RBF kernel, we have a new kernel termed graph

spectral Gaussian (GSG) kernel:

$$\mathcal{K}(x, x'; l) = exp(-\frac{||U^\top x - U^\top x'||^2}{2l^2}),\tag{5}$$

where $l$ is a hyperparameter corresponding to the length-scale of the RBF kernel. Mercer kernels are essential for Gaussian Processes (GPs) as they ensure valid covariance matrices and enable implicit high-dimensional data mapping. Additionally, they offer computational benefits through the "kernel trick" in expansive spaces. Therefore, we analyze if the proposed kernel is a valid Mercer kernel.

**Theorem 4.1.** *GSG is a valid Mercer Kernel for GP.*

*Proof.* The kernel in Equation 5 can be transformed as follows:

$$\mathcal{K}(x, x'; l) = exp(-\frac{||U^\top x - U^\top x'||^2}{2l^2}) = exp(-\frac{[(U^\top x)^\top U^\top x + (U^\top x')^\top U^\top x' - 2(U^\top x)^\top U^\top x']}{2l^2})$$

$$= exp(-\frac{[x^\top U U^\top x + x'^\top U U^\top x' - 2x^\top U U^\top x']}{2l^2}) = exp(-\frac{||x - x'||^2}{2l^2}).$$

Hence, $\mathcal{K}(x, x'; l)$ can be considered as being equivalent to the RBF kernel, which is widely recognized as a valid Mercer kernel. $\square$

Next, we set up a Gaussian process (GP) with GSG kernel to realize Equation 2. The purpose of this GP is to estimate the expected influence spread $\phi$ of the provided seed set $s$ evaluated by simulation.

$$GP : s \rightarrow \phi(s).\tag{6}$$

## 4.3 DATA ACQUISITION

In order to prepare the data for training Equation 6, it is necessary to obtain a series of pairings $(s_i, \phi(s_i))$. The acquisition of each individual $\phi(s_i)$ necessitates a simulation, which is an expensive process. The objective is to choose more representative node sets, aiming for maximum diversity in order to minimize variance.

The GP requires initialization and iterative training, both utilizing sampling techniques. Initialization requires sampling multiple data points, while each iteration selects a new data point from a fresh sample set by maximizing an acquisition function. This acquisition function leverages the GP posterior to balance exploration and exploitation. Due to the discrete property of the graph data, traditional sampling methods commonly employed, such as the Sobol sequence Sobol' (1967), do not fit the influence maximization problem. As a replacement, we propose a graph stratified sampling (GSS), which clusters the candidate and sample uniformly from each group. Specifically, GSS performs clustering over graph Fourier signals of candidate sources (Equation 4), and samples equal-size candidates from each cluster.

**Theorem 4.2.** *GSS has a lower variance than random sampling.*

*Proof.* Simple random sampling randomly draws $m$ samples from the entire population. The variance of its mean estimator is:

$$\text{Var}(\bar{Y}_{\text{rs}}) = \text{Var}\left(\frac{\sum_{i=1}^m Y_i}{m}\right) = \frac{1}{m^2}\text{Var}\left(\sum_{i=1}^m Y_i\right) = \frac{\sigma^2}{m},$$

where $\sigma^2 = \text{Var}(Y_i)$ is the population variance. To set up GSS, we divide all candidates into $\kappa$ non-overlapping equal-sized groups based on similarity. $N$ is the population, and $N_i$ is the population in $i$-th group. From the $i^{th}$ group, $m_i$ samples are drawn, with a total of $m = m_1 + m_2 + \cdots + m_\kappa$ samples. The variance of this GSS mean estimator is given by:

$$\text{Var}(\bar{Y}_{\text{gss}}) = \text{Var}\left(\sum_{i=i}^\kappa \bar{Y}_i\right) = \sum_{i=1}^\kappa \left(\frac{N_i}{N}\right)^2 \frac{\sigma_i^2}{m_i},\tag{7}$$

where $\sigma_i$ is the sample mean of the $i^{th}$ group. To demonstrate the variance reduction of GSS compared to simple random sampling, we compare $\text{Var}(\bar{Y}_{\text{gss}})$ and $\text{Var}(\bar{Y}_{\text{rs}})$. Note that the within-group similarity exists, so the variances within each group are smaller than the overall population variance, i.e., $\forall i, \sigma^2 \geq \sigma_i^2$. In addition, the sample size of each group is the same (i.e., $m_1 = m_2 = \ldots = m_c = \tilde{m}$, and $\kappa \cdot \tilde{m} = m$), the size of each group is the same ($\frac{N}{N_i} = \kappa$). So:

$$\text{Var}(\bar{Y}_{\text{gss}}) = \sum_{i=1}^\kappa \left(\frac{1}{\kappa}\right)^2 \frac{\sigma_i^2}{\tilde{m}} \leq \sum_{i=1}^\kappa \left(\frac{1}{\kappa}\right)^2 \frac{\sigma^2}{\tilde{m}} = \frac{1}{\kappa}\frac{\sigma_i^2}{\tilde{m}} = \text{Var}(\bar{Y}_{\text{rs}}).$$

$\square$

The underlying assumption is that GSS clusters similar items within each group, thereby reducing within-group variance and, consequently, the estimator's overall variance. This would aid Bayesian Optimization in minimizing overall variance and drawing precise conclusions about actual sources. Note that the expected sample mean by GSS is identical to the sample mean by random sampling, which is the population mean. Consequently, a decrease in variance reduces inference errors.

Expected improvement (EI) is used to estimate the potential improvement of samples over the current best observation. Suppose the model clusters all candidates into $\gamma$ groups $C = \{c_1, c_2, \ldots, c_\gamma\}$, one set is sampled from each group such that $s_i \sim c_i, \forall i \in [1, \gamma]$. We optimize EI over the sample set $[s_1, s_2, \ldots, s_\gamma]$ such that:

$$\tilde{s}^* = \underset{\tilde{s}_i \in [\tilde{s}_1, \tilde{s}_2, \ldots, \tilde{s}_\gamma]}{\arg\max} \text{EI}(\tilde{s}_i) = \underset{\tilde{s}_i}{\arg\max} \mathbb{E}[\delta(s_i, s+) \cdot I(s_i)],$$

where $s+$ is the best set so far, $s_i$ is the node set that corresponds to the graph Fourier transform signal $\tilde{s}_i$, and $\delta(s, s+) = f(s; o^*) - f(s+; o^*)$. $I(s_i)$ is an indicator function that equals to 1 when $f(s_i; o^*) > f(s+; o^*)$ and 0 when otherwise. Although the search space in our problem is finite, enumerating all node sets in each iteration violates our principle of efficiency. Thus, we strategically sample a few sets with GSS and use EI to pick the maximum. For the initial node sets and the node set in each iteration, we query the true value of the objective function as the expected influence spread $\phi$. The evaluation is achieved by simulations based on the given diffusion models such as IC or LT.

### 4.4 ALGORITHM

**BOIM** is demonstrated in Algorithm 1. Initiated with a graph $G$, a given diffusion model $d$, time step $t$, IM budget $k$, BO budget $\beta$, sample size $\gamma$, and filter threshold $\delta$, it aims to produce an $k$-sized node set $\Omega$ that maximize the expected influence spread $\phi$. The algorithm selects the top $a$ nodes based on degree centrality as the candidate pool. A graph Fourier transform is applied on all $k$-size subsets of the candidate pool that pass the distance filter (lines 3-11). These transformed sets are clustered into $c$ groups for later stratified sampling (line 12). One graph Fourier transform signal is randomly sampled from each cluster and evaluated by simulations given the corresponding node set as the seeds (line 16). The $c$ pairs of Fourier representation of sources and influence spread are used to train the GP model as an initialization (line 13-19). In each following iteration, a new group of data points is sampled by GSS, and one of them is picked by the EI acquisition function. After evaluation, the GP model is updated with the new signal-spread pair and the process repeats until convergence or the iteration budget is used up (line 20-27). After that, all candidate sets are evaluated with the trained GP model to find the spread-maximizing seed set $\Omega$ (line 28).

### 4.5 TIME COMPLEXITY

We analyze the time complexity of **BOIM** based on Algorithm 1. Selecting $a$ nodes with the highest degree centralities (line 2) is $\mathcal{O}(|V| + |E|) = \mathcal{O}(N^2)$ using BFS traversal. This complexity can be further reduced to $\mathcal{O}(N)$ for sparse graphs. Calculating the graph Fourier transform operator is $\mathcal{O}(N^3)$ Merris (1994). Generating all $k$-sized node sets from the $a$ candidate nodes (line 3) requires $\mathcal{O}(a^k)$. Looping through all combinations has a time complexity of $\mathcal{O}(a^k)$, and the operations inside the loop are multiplications between $1 * N$ vectors and $N * N$ matrices, which are $\mathcal{O}(N^2)$. Thus, the time complexity for the whole block (line 2-11) is $\mathcal{O}(a^k N^2)$. Clustering the graph Fourier signals is $\mathcal{O}(a^k)$. The operations above are the preparations for the GP training and have a combined time complexity of

$$\mathcal{O}(N^2 + N^3 + a^k + a^k N^2 + a^k) = \mathcal{O}(N^3),$$

when $a \ll N$ and $k$ is a very small integer. Breaking down the GP training process and the final prediction (line 13 - 28), we get all the operations carried out. There are $(\gamma + \gamma(\beta - \gamma))$ samplings from the cluster, $\beta$ simulations to evaluate the expected influence spread, $(\beta - \gamma + 1)$ rounds of GP model training, and 1 evaluation for each candidate node sets using the trained GP. Assuming each simulation takes a long but constant time $T$, the time complexities of sampling, simulation, GP training, and GP evaluation are $\mathcal{O}(1), \mathcal{O}(T), \mathcal{O}(|\beta|^3)$, and $\mathcal{O}(1)$ Rasmussen et al. (2006), respectively. Thus, the time complexity for the whole training and predicting period is $\mathcal{O}(\gamma + \beta\gamma - \gamma^2 + \beta T + (\beta - \gamma + 1)|\beta|^3 + a^k) = \mathcal{O}(T)$ since $\beta$ and $\gamma$ are constants, and $a^k$ is

ignorable comparing to the problem size $N$. The overall time complexity of **BOIM** is

$$\mathcal{O}(N^2 + N^3 + a^k + a^k N^2 + a^k + \gamma + \beta\gamma - \gamma^2 + \beta T + (\beta - \gamma + 1)|\beta|^3 + a^k) = \mathcal{O}(N^3 + T) \quad (8)$$

where $N$ is the graph size, $k$ is the budget for seed selection, and $T$ is the time spent on evaluating the influence spread of one node set. Comparatively, the original simulation-based greedy method (GRD) Kempe et al. (2003) performs $N$ simulations to find the first seed. For each following seed, the number of simulations reduces by 1. Thus, the total number of simulations is $N + (N - 1) + \cdots + (N - k + 1) = kN - (k^2 - k)/2$ and the time complexity is $\mathcal{O}(kNT)$. The time complexity for the fastest simulation-based IM, which is Sobol Influence Maximization Zhang & Chen (2023) (SIM), is $\mathcal{O}(M)$ where $M$ is a proxy-based IM algorithm that combines with the following simulations. It is claimed that SIM combined with Degree Discount (SIM-DD) could provide a good enough solution. Thus, the time complexity can be regarded as $\mathcal{O}(k \log N)$. However, this time complexity considers the evaluation time neglectable. As the evaluation is #$P$-hard, which is at least as hard as $NP$ problems, we need to consider the evaluation time. Therefore, the actual time complexity for SIM combined with degree discount is $\mathcal{O}(k \log N + 2^k T)$. Assuming that the complexity of $T$ dominates $N^3$, we can put the time complexities of the three methods above together:

- **BOIM**: $\mathcal{O}(T)$
- GRD: $\mathcal{O}(kNT)$
- SIM-DD: $\mathcal{O}(2^k T)$

We can observe that **BOIM** requires significantly fewer simulations than GRD and SIM-DD. Runtime analysis shows that **BOIM** is about 1.5 times faster than SIM-DD, and about 17 to 22 times faster than GRD. Details are discussed in the experiment section.

---

**Algorithm 1 BOIM with GSS**

**Input:** Graph $G$, IM budget $k$, diffusion model $d$, time step $t$, sample size $\gamma$, BO budget $\beta$, distance threshold $\delta$
**Output:** A $k$-sized node set $\Omega$

1: set $S \leftarrow \emptyset$, set $\Phi \leftarrow \emptyset$, set $\tilde{S} \leftarrow \emptyset$
2: $pool \leftarrow$ top $a$ nodes by degree centrality
3: **for** $k$-size node set $s \subset pool$ **do**
4:     **if** Shortest-path between nodes in $s \geq \delta$ **then**
5:         $S \leftarrow S + s$
6:     **end if**
7: **end for**
8: **for** $s \in S$ **do**
9:     $\tilde{s} \leftarrow U^\top s$ as in Eq. 4
10:     $\tilde{S} \leftarrow \tilde{S} + \tilde{s}$
11: **end for**
12: cluster $\tilde{S}$ into $b$ groups: $C = \{c_1, c_2, \ldots, c_\gamma\}$
13: sample one set from each set group, $\{\tilde{s}_i\}_{i=1}^{\gamma} \sim c_i$
14: **for** $\tilde{s}_i \in [\tilde{s}_1, \tilde{s}_2, ..., \tilde{s}_\gamma]$ **do**
15:     $s_i \leftarrow S[loc(\tilde{s}_i)]$ where $loc(\cdot)$ is the index of $\cdot$ in $\tilde{S}$
16:     $\phi_i \leftarrow$ simulate $d$ with $t$ on $G$ with $s_i$ as sources
17:     $\Phi \leftarrow \Phi + (\tilde{s}_i, \phi_i)$
18: **end for**
19: train GP (as surrogate) with $\Phi$: $\tilde{s} \xrightarrow{GP} \phi$
20: **while** $z \neq 0$ ($z = \beta - \gamma$) **do**
21:     sample one set from each set group, $\{\tilde{s}_i\}_{i=1}^{\gamma} \sim c_i$
22:     $\tilde{s}^* \leftarrow \arg\max_{\tilde{s}_{ij}} \text{EI}(\tilde{s}_{ij}), \forall \tilde{s}_{ij} \in [\tilde{s}_{11}, \tilde{s}_{12}, ..., \tilde{s}_{b\gamma}]$
23:     $s^* \leftarrow S[loc(\tilde{s}^*)]$
24:     $\phi^* \leftarrow$ simulate $d$ within $t$ on $G$ with $s^*$ as sources
25:     $\Phi \leftarrow \Phi + (\tilde{s}^*, \phi^*)$ and re-train GP with $\Phi$
26:     $z \leftarrow z - 1$
27: **end while**
28: Find optimal $\tilde{s}$ with GP: $i = \arg\max_i \text{GP}(\tilde{s}_i)$
29: $\Omega = s_i \in S$

---

## 5 EXPERIMENT

**Configurations.** We evaluate **BOIM** on synthetic and real-world datasets. The experiments are carried out on a server with AMD EPYC 7302P CPU with 32GB RAM. Simulations are performed by NDLib Rossetti et al. (2018), an open-source toolkit for diffusion dynamics. The Bayesian optimization paradigm is implemented by BOTorch Balandat et al. (2020) and GPyTorch Gardner et al. (2018). Our code for the experiments is available in the supplementary materials. We adopt IC and LT as diffusion models to evaluate **BOIM**'s performance across multiple diffusion models. Each seed set is evaluated by 100 simulation rounds. The Bayesian optimization paradigm includes 200 iterations to train the final model. $d_s$ is set to be larger than or equal to 2.

**Datasets.** Four datasets are employed to evaluate the effectiveness and the efficiency of **BOIM**. Three real-world datasets, namely Cora, CiteSeer, and PubMed Yang et al. (2016), reproduce the complex social network structure. Since the IM problem is traditionally studied on connected graphs, we take the largest connected component of these graphs as the studied network. A synthetic *connected Watts-Strogatz small-world graphs* (SW) is generated using NetworkX to represent pseudo social networks. It is generated with 3000 nodes for effectiveness evaluation. We also use

SW graphs with sizes ranging from 1000 to 5000 for runtime analysis. For all graphs, each edge is uniformly and randomly assigned a weight between 0.40 and 0.80 for the IC model as the activation probabilities, and each node is assigned a threshold between 0.01 to 0.20 for the LT model.

**Baselines.** We select popular IM algorithms as our baselines: (1) *Simulation-based greedy algorithm (GRD)* Kempe et al. (2003): This approach adds the node with the highest marginal influence spread to the seed set in each iteration. The marginal influence spread is determined by averaging the results of 1000 simulation rounds. (2) *Degree Centrality (DEG)*: This method selects the $k$ nodes with the highest degree centrality. After each selection, the chosen seed is removed from the graph, and the degrees of the remaining nodes are updated, known as SingleDiscount Chen et al. (2009). (3) *Eigenvector Centrality (EIG)*: This technique picks the first $k$ nodes ranked by their eigenvector centrality. (4) *Degree discount (DD)* Chen et al. (2009): Similar to DEG, but the degree of each candidate node is discounted based on the likelihood of its activation. (5) *Sigma* Yan et al. (2019): This method estimates the spreading power of nodes using $\sum_t I \cdot A^t$, where $I$ is a unit column vector. Nodes are selected based on this estimation in a greedy manner. (6) *Pi* Zhang et al. (2022): Nodes are chosen based on their estimated spreading powers, calculated using $I \cdot \left( J - \prod_{r=1}(1 - A^r) \right)$, where $J$ is an all-one matrix and $\prod$ represents the element-wise product of matrices. (7) *Sobol degree discount (SIM-DD)* Zhang & Chen (2023): Initially, the DD algorithm selects $2k$ nodes, which are then pruned based on their Sobol total indices.

## 5.1 RESULTS

The empirical study generally consists of three major parts: (1) **effectiveness verification**: demonstrate the effectiveness of the proposed algorithm against current IM algorithms. (1) **ablation study**: we compare **BOIM** with two variants to evaluate the utility of our proposed GSG and GSS. (3) **runtime analysis**: compare the **BOIM** with the baseline IM methods to evaluate its time efficiency.

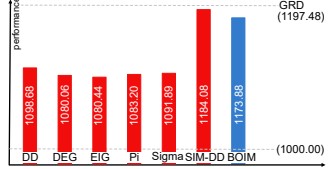

FIGURE 2: Results on Cora with IC.

### 5.1.1 EFFECTIVENESS VERIFICATION

To demonstrate the effectiveness of **BOIM**, we compare it with the baselines on the four datasets. First, experiments are carried out on the largest component of the Cora graph. The graph size is 2485. Under the IC model, the simulation-based greedy algorithm (GRD) takes 10678.53 seconds to find a 3-seed set. Comparatively, **BOIM** runs for 624.58 seconds and SIM-DD takes 983.42 seconds. We can conclude that **BOIM** accelerates the simulation-based method, shrinking the running time to $\frac{1}{17}$, and

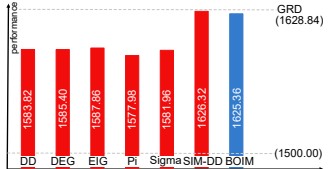

FIGURE 3: Results on Cora with LT.

the acceleration effect is greater than that of SIM-DD. The IM performances of the seven methods are compared with GRD, the performance SOTA, in Figure 2. **BOIM** achieves the same performance level with GRD and SIM-DD. Experiment with the LT model shows a similar result. Figure 3 demonstrates that **BOIM** achieves an influence spread of 1625.36 within 941.33 seconds while a similar performance of 1628.84 takes GRD 20824.73 seconds.

We also compare the performance of **BOIM** with the baselines on the other three graphs. Each IM algorithm generates 3 seeds to maximize the expected influence spread, which is measured by the mean and standard deviation of 100 simulations to remove the systematic uncertainty. The detailed results under the IC model and the LT model are presented in Table 1 and Table 2, respectively.

TABLE 1: Results with IC model.

| Methods | SW(3000) | Cora(2485) | CiteSeer(3679) | PubMed(44324) |
|---|---|---|---|---|
| DD | $1638.92 \pm 45.59$ | $1098.68 \pm 8.27$ | $617.16 \pm 10.39$ | $10144.32 \pm 32.40$ |
| DEG | $1599.00 \pm 45.36$ | $1080.06 \pm 19.47$ | $604.06 \pm 9.38$ | $9139.68 \pm 49.86$ |
| EIG | $1625.00 \pm 32.01$ | $1080.44 \pm 10.32$ | $414.88 \pm 4.14$ | $3817.02 \pm 37.95$ |
| Pi | $1711.90 \pm 30.54$ | $1083.20 \pm 14.21$ | $607.08 \pm 9.01$ | $9166.20 \pm 39.35$ |
| Sigma | $1380.66 \pm 27.77$ | $1091.90 \pm 8.91$ | $406.80 \pm 11.71$ | $9886.76 \pm 55.98$ |
| SIM+DD | $1752.96 \pm 23.78$ | $1184.08 \pm 14.95$ | $669.92 \pm 4.27$ | $10182.76 \pm 69.69$ |
| BOIM | $1743.30 \pm 48.99$ | $1173.88 \pm 33.55$ | $673.62 \pm 17.71$ | $10121.50 \pm 171.64$ |

viation of 100 simulations to remove the systematic uncertainty. The detailed results under the IC model and the LT model are presented in Table 1 and Table 2, respectively.

We can observe that **BOIM** achieves significantly higher influence spread than the proxy-based methods. Its performance is competitive, sometimes higher, compared to the other simulation-based method SIM-DD.

### 5.1.2 ABLATION STUDY

Table 3 compares the performance of **BOIM** against three ablated versions: using all candidate sets without filtering, using random sampling (RS) instead of GSS, and using an RBF kernel instead of the proposed GSG. Column (+GSG)

TABLE 2: Performance with LT model.

| Methods | SW(3000) | Cora(2485) | CiteSeer(3679) | PubMed(44324) |
|---|---|---|---|---|
| DD | 622.78 ± 23.13 | 1583.82 ± 12.02 | 910.76 ± 16.28 | 7283.24 ± 311.22 |
| DEG | 633.12 ± 29.85 | 1585.40 ± 10.88 | 911.84 ± 11.20 | 6778.99 ± 386.63 |
| EIG | 587.36 ± 23.95 | 1587.86 ± 17.51 | 578.2 ± 4.34 | 3126.58 ± 206.82 |
| SIM+DD | 660.88 ± 53.57 | 1626.32 ± 7.95 | 997.58 ± 16.70 | 7564.17 ± 214.76 |
| BOIM | 690.42 ± 44.09 | 1625.36 ± 104.34 | 988.64 ± 38.92 | 7591.63 ± 424.75 |

shows the percentage increase in influence spread after substituting the RBF kernel with the GSG kernel. Column (+GSS) demonstrates the further performance increase after combining GSS with GSG. Column (+Filter) represents the performance increase percentage from the distance filter given that GSG and GSS have already been applied.

We can observe that RS + GSG always outperforms RS + RBF. More specifically, GSG brings performance enhancement ranging from $1.95\% - 5.90\%$. This shows the benefits of the graph spectral Gaussian ker-

TABLE 3: Ablation tests under IC with RS+RBF without distance filter.

| Methods | SW(3000) | Cora(2485) | CiteSeer(3679) | PubMed(44324) |
|---|---|---|---|---|
| RS+RBF | 1616.20 ± 85.45 | 1036.42 ± 50.83 | 561.78 ± 42.75 | 9096.28 ± 722.70 |
| (+GSG) | +1.95% | +4.58% | +5.90% | |
| RS+GSG | 1647.76 ± 85.93 | 1083.88 ± 40.41 | 594.94 ± 17.85 | 9300.34 ± 497.48 |
| (+GSS) | +3.69% | +5.51% | +5.10% | |
| GSS+GSG | 1708.60 ± 61.11 | 1143.56 ± 30.63 | 625.26 ± 46.89 | 10028.04 ± 310.82 |
| (+Filter) | +2.03% | +2.65% | +7.73% | |
| GSS+GSG+Filter | 1743.30 ± 48.99 | 1173.88 ± 33.55 | 673.62 ± 17.71 | 10121.50 ± 171.64 |

nel for effective adaptation to the graph-structured data and the influence maximization problem. It is also demonstrated that GSS + GSG outperforms RS + GSG on all four datasets. The increase in influence spread ranges from $3.69\% - 7.82\%$ compared to before GSS is applied. This shows the benefits of graph stratified sampling for uniform data acquisition. It explores the search space better than random sampling. The distance filter also bounces the performance by $0.93\%$ to $7.73\%$. In sum, our ablation study verifies that the proposed components of GSS, GSG, and distance filter provide significant gains over variants without those techniques. GSG kernel consistently assists in graph-structured data adaptation, fulfilling the smoothness assumption. Graph stratified sampling is crucial for handling more complex search spaces. On simpler and smaller graphs like SW, GSS provides diminishing benefits. Distance filter helps with sampling more efficiently and thus benefits the GP training.

### 5.1.3 RUNTIME ANALYSIS

As expected, the two simulation-based methods take significantly longer than the proxy-based methods, and their scalability

TABLE 4: Runtime (in seconds) comparison on SW graphs.

| Methods | 1000 | 2000 | 3000 | 4000 | 5000 |
|---|---|---|---|---|---|
| DD | 0.01 ± 0.00 | 0.04 ± 0.00 | 0.07 ± 0.00 | 0.11 ± 0.01 | 0.17 ± 0.00 |
| Pi | 0.23 ± 0.00 | 0.70 ± 0.00 | 1.60 ± 0.01 | 2.78 ± 0.02 | 4.34 ± 0.02 |
| Sigma | 0.27 ± 0.00 | 1.09 ± 0.15 | 2.02 ± 0.02 | 3.46 ± 0.27 | 5.02 ± 0.30 |
| SIM+DD | 182.85 ± 1.49 | 596.76 ± 1.96 | 1303.18 ± 9.37 | 2292.32 ± 8.77 | 3618.68 ± 28.73 |
| BOIM | 118.43 ± 0.46 | 380.13 ± 0.81 | 853.20 ± 2.91 | 1487.52 ± 10.74 | 2322.41 ± 19.77 |

is worse. This is due to the high complexity of the simulations. When compared with each other, **BOIM** shows higher efficiency than SIM-DD. The runtimes of the two algorithms are closest to each other when the graph size $N = 1000$. **BOIM** takes 118.43 seconds and SIM-DD takes 182.82 seconds, leaving the time difference at 64.39 seconds. We can also say that **BOIM**'s runtime is $64.78\%$ of SIM-DD's runtime. As the graph grows from 1000 to 5000, this time difference enlarges to 1296.27 seconds. And when the graph size $N = 5000$, **BOIM** takes $64.18\%$ of SIM–DD's runtime. Generally speaking, the two algorithms have similar scalability, but **BOIM** runs about 1.5 times faster than SIM-DD.

## 6 CONCLUSION

We present an efficient simulation-based method **BOIM** for influence maximization. Bayesian optimization is adopted to reduce the number of simulations and uncover the black-box relationship between the seed sets and their corresponding influence spread. We theoretically prove that GSG, a graph-level kernel for the Gaussian process, is a valid Mercer. It is also proven that GSS, which is graph stratified sampling based on the clustering of graph Fourier signals, reduces variance better than random sampling. **BOIM** demonstrates competitive performance in empirical experiments conducted on synthetic and real-world datasets. Ablation studies show that GSG, GSS, and the distance filter help with the performance of **BOIM**.

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
