# OpenReview forum: "Accelerating Simulation-Based Influence Maximization via Bayesian Optimization"
_ICLR.cc/2024/Conference — Submitted to ICLR 2024_

### Official Review · Reviewer_sPAd · 2023-10-22

**Soundness:** 2 fair
**Presentation:** 1 poor
**Contribution:** 2 fair
**Rating:** 3
**Confidence:** 4

**Summary:**

This paper proposes a Bayesian optimization framework for reducing the number of seed sets for which influence spread needs to be simulated to find a high quality seed set for the influence maximization problem. A novel kernel and sampling method are proposed to adequately apply Bayesian optimization to the influence maximization problem. Results show comparable seed set quality with state-of-the-art methods while running in a small fraction of the time.

**Strengths:**

* The overall approach is straight-forward and well motivated. This influence maximization problem is a good candidate for Bayesian optimization.
* The results show the efficacy of the proposed method.
* Figure 1 is very helpful in understanding the proposed method.

**Weaknesses:**

* The proposed GSG kernel has the same issue (described in the first paragraph of Section 4.2) as any appropriate kernel that operates on the binary representations of seed sets. This is shown in the proof of Theorem 4.1 and is due to the fact that the columns of $U$ are orthonormal.
* Section 3 seems unnecessary -- an over-explanation of the influence maximization problem.
* The presentation of the results could be improved quite a bit. The histograms and tables are quite small and do not advocate for the proposed method in a clear and coherent manner. Some space could be created by cutting some of the unnecessary material earlier in the paper.
* The time complexity results are important, but the derivation could probably be moved to supplementary material or an appendix.

**Questions:**

* How is the GSG different than just using an RBF kernel? If GSG is equivalent to RBF, how is GSG performing better than RBF (see Table 3)? If GSG is not equivalent to RBF, is there something wrong with the theoretical analysis in Section 4?
* Should the kernel be different for different types of diffusion models?
* The five cheaper baselines perform quite well in much less time than the proposed method. What is the time-performance trade-off to be made by a potential user?

---

### Official Review · Reviewer_CMuJ · 2023-10-31

**Soundness:** 2 fair
**Presentation:** 3 good
**Contribution:** 1 poor
**Rating:** 3
**Confidence:** 4

**Summary:**

This manuscript introduces a Bayesian Optimization Influence Maximization (BOIM) framework that employs Bayesian optimization to minimize the number of required simulations. The BOIM approach utilizes a Gaussian Process (GP) as a surrogate function for modeling the complex interaction between seed sets and influence spread. A specialized kernel for graph-level Bayesian optimization is introduced in GP, and stratified sampling is employed to ensure uniform instance distribution. Extensive experiments are conducted.

**Strengths:**

1. Reproducible as the source code is attached.
2. Writing is concise and clear.
3. This work introduces Bayesian Optimization into the field of IM, which may provide potentially useful insights.

**Weaknesses:**

1. Some important references are missing. For Influence Maximization, the state-of-the-art RIS-based papers are not reviewed in this manuscript, like IMM [1], OPIM [2] and SUBSIM [3].
2. Baselines are outdated. In the field of IM, there already exist many scalable methods (IMM, OPIM, SUBSIM), which are not compared in this manuscript. So the claim of "approximately 1.5 times faster than the most efficient existing algorithm" is misleading.
3. The proposed method is neither scalable (with time complexity $O(N^3)$) nor theoretically guaranteed (from the perspective of approximation ratio).
4. From the perspective of performance, I do not see BOIM outperforms SIM-DD much.
5. Runtime analysis on the real-world datasets is missing, which I think is more important than that of the sythetic ones.
6. Some assumptions are questionable (See Questions).

[1] Tang, Y., Shi, Y., & Xiao, X. (2015, May). Influence maximization in near-linear time: A martingale approach. In Proceedings of the 2015 ACM SIGMOD international conference on management of data (pp. 1539-1554).

[2] Tang, J., Tang, X., Xiao, X., & Yuan, J. (2018, May). Online processing algorithms for influence maximization. In Proceedings of the 2018 International Conference on Management of Data (pp. 991-1005).

[3] Guo, Q., Wang, S., Wei, Z., & Chen, M. (2020, June). Influence maximization revisited: Efficient reverse reachable set generation with bound tightened. In Proceedings of the 2020 ACM SIGMOD international conference on management of data (pp. 2167-2181).

**Questions:**

1. Why do you utilize a Gaussian Process (GP) model to function as a surrogate for predicting the influence spread? Any theoretical or empirical support?
2. In the time complexity analysis part, why do you assume that $T$ dominates $N^3$?
3. In section 4.5, BOIM is faster than SIM-DD by a factor of $2^k$. But the experiments state that BOIM is only about 1.5 times faster. Why is that?

---

### Official Review · Reviewer_U1hZ · 2023-10-31

**Soundness:** 2 fair
**Presentation:** 2 fair
**Contribution:** 2 fair
**Rating:** 3
**Confidence:** 4

**Summary:**

The paper discusses the challenges in Influence Maximization (IM). The authors introduce a Bayesian Optimization Influence Maximization (BOIM) framework, using a Gaussian Process to reduce simulation needs while accurately modeling seed set influences. Initial tests suggest BOIM offers both computational efficiency and high approximation accuracy.

**Strengths:**

1. Introduces a principled Bayesian optimization framework for simulation-based IM to improve efficiency.
2. Proposes novel graph-based kernel and sampling methods tailored for the IM problem.

**Weaknesses:**

1. There is key issue in the assumption of the algorithm design. In most real-world applications, the important nodes are in most cases O(N) or at least O(poly(N)) like sqrt(N) etc. And the number of seeds to be selected in real marketing applications are usually at least few hundreds for like graph with 10k nodes and much more in real-world networks with millions or billions of nodes. The O(a^k) algorithm is not a reasonable algorithm to solve the problem at all.
2. The proposed algorithm is highly inefficient and there are issues in the time complexity analysis of the proposed method and competing methods. For simulation based method, though the exact evaluation is #P-hard, sampling based method with a few thousand samples can converge to a very high accuracy as shown in previous work. In an efficient implementation, T = O(M) instead of O(2^k). And the dominant term of the proposed algorithm is O(N^3). The issue exists in the analysis of methods from the literature.
3. The authors fail to include comparison in both time complexity analysis and empirical evaluation to the state-of-art near-linear time algorithm with approximation guarantee like Influence Maximization in Near-Linear Time: A Martingale Approach and many more recent algorithms. The propose method also relies on slow simulation to sample value of influence function and assumes either IC and LT model which can both be solved much more efficient with existing methods.
4. The proposed graph kernel is not well justified as why the proposed kernel is a better choice. Either theoretical or empirical analysis will help. Also the authors should at least properly define the variance in Theorem 4.2. Moreover, it is not clear how it helps the algorithms overall.
5. The empirical evaluation is not convincing. First, the authors should clearly present the experiment setting like graph statistics, number of seed nodes. Second, the number of seed selected is too small (3 is used in experiment), which is not reasonable in any real-world applications. Third, the greedy method takes 10678s to select only three seeds on a 2k nodes graph.  Even the most naive implementation with optimization like CELF should not take this long. Since it only requires 3 seeds * 100 rounds *2000^2 (N^2 assume dense graph which is far higher than in real case) computation only. Forth, as mentioned above, the authors should include comparison to the near-linear time algorithms in comparison. the IMM algorithm uses ~100 seconds to find 50 seeds in a network of 40M nodes (Influence Maximization in Near-Linear Time: A Martingale Approach and many more recent algorithms).

**Questions:**

Please see above Weaknesses section.

---

### Official Review · Reviewer_sqPq · 2023-11-01

**Soundness:** 3 good
**Presentation:** 3 good
**Contribution:** 3 good
**Rating:** 6
**Confidence:** 4

**Summary:**

The paper proposes a efficient simulation-based method for influence maximization (IM) using Bayesian Optimization (BO) called BOIM. The paper proposes a novel kernel for graph as well as a new sampling technique (GSS) to exploit the idiosyncratic structure of graph data, leading to superior performance than baseline methods. The paper has also performed theoretical and empirical evaluation of the proposed method.

**Strengths:**

- The paper studies an important problem (information maximization) and has applied Bayesian Optimization (BO) on this problem with novel kernel and sampling techniques.
- The paper is in general very well-written and easy to follow.
- The proposed method is not only displaying good optimization performance, but also cheap in computation relative to simulation-based methods

**Weaknesses:**

- My main concern is that the performance improvement, though generally better, is not particularly too significant, not to mention that those proxy-based method achieves also pretty good IM results while using only a negligible amount of time compared to BOIM (or other simulation-based method in general)
- Other choices of graph kernel are not considered and experimented with such as random walk or Graphlet kernel? There are probably easy tricks to turn them into valid GP kernels.
- Despite the time reduction introduced by BOIM, proxy-based methods are still substantially cheaper. Would it be possible to use proxy-based methods as heuristics to seed BOIM or other zero-order optimization method (e.g., CMA-ES).
- While the author has shown that GSS has theoretically lower variance, it’d be nice to compare against with random sampling and check empirically how well it performs.
- Results presentation can be improved. For example, in Figure 2 and 3, the y-axis is labeled as “performance” which is ambiguous, and the runtime is not represented in those figure. A scatter plot with x/y axes being runtime/performance could help the reader better understand and interpret the results. Best results in tables can also be highlighted.

Minor:

- Typo in Section 2: “Mockus (1998) and has since become…” → “Mockus (1998) has since become…”

**Questions:**

See questions raised in weakness above.

---

### Meta-Review · Area_Chair_jLGN · 2023-12-05

**Metareview:**

This paper presents a method for speeding up simulation-based influence maximization using Bayesian optimization, so that it is expected to minimize the number of required simulations.  The method is referred to as BOIM where a novel kernel for  graph as well as a new sampling technique is proposed. In general, the topic itself is timely and important and the paper is well written. However, there are a few critical concerns raised by reviewers. Most of reviewers pointed out that   the proposed method is not indeed scalable (see their detailed comments). It is also criticized that the empirical comparison is not convincing either. The authors did not address the reviewers’ concerns, without providing the rebuttal. So, all those concerns remained. We feel that the paper is not ready for being published on ICLR in its current version. Therefore, the paper is not recommended for acceptance in its current form. I hope authors found the review comments informative and can improve their paper by addressing these carefully in future submissions.

**Justification For Why Not Higher Score:**

Both the algorithm and empirical comparison are not convincing.

**Justification For Why Not Lower Score:**

N/A

---

### Decision · Program_Chairs · 2024-01-16

Reject